# Numerical study on advective fog formation and its characteristic associated with cold water upwelling

Soon-Young Park[1], Jung-Woo Yoo[1], Sang-Keun Song[2], Cheol-Hee Kim[3], Soon-Hwan Lee[4]*

1 Institute of Environmental Studies, Pusan National University, Busan, Republic of Korea, 2 Department of Earth and Marine Sciences, Jeju National University, Jeju, Republic of Korea, 3 Department of Atmospheric Sciences, Pusan National University, Busan, Republic of Korea, 4 Department of Earth Science Education, Pusan National University, Busan, Republic of Korea

* withshlee@pusan.ac.kr

## Abstract

Recent rapid industrial development in the Korean Peninsula has increased the impacts of meteorological disasters on marine and coastal environments. In particular, marine fog driven by summer cold water masses can inhibit transport and aviation; yet a lack of observational data hinders our understanding of this phenomena. The present study aimed to analyze the differences in cold water mass formation according to sea surface temperature (SST) resolution and its effects on the occurrence and distribution of sea fog over the Korean Peninsula from June 23–July 1, 2016, according to the Weather Research and Forecasting model. Data from the Final Operational Model Global Tropospheric Analyses were provided at 1˚ and 0.25˚ resolutions and NOAA real-time global SST (RTG-SST) data were provided at 0.083˚. While conventional analyses have used initial SST distributions throughout the entire simulation period, small-scale, rapidly developing oceanic phenomena (e.g., cold water masses) lasting for several days act as an important mediating factor between the lower atmosphere and sea. RTG-SST was successful at identifying fog presence and maintained the most extensive horizontal distribution of cold water masses. In addition, it was confirmed that the difference in SST resolution led to varying sizes and strengths of the warm pools that provided water vapor from the open sea area to the atmosphere. On examining the horizontal water vapor transport and the vertical structure of the generated sea fog using the RTG-SST, water vapors were found to be continuously introduced by the southwesterly winds from June 29 to 30, creating a fog event throughout June 30. Accordingly, high-resolution SST data must be input into numerical models whenever possible. It is expected that the findings of this study can contribute to the reduction of ship accidents via the accurate simulation of sea fog.

**Data Availability Statement:** All simulations and observed data are available from the Zenodo database (https://doi.org/10.5281/zenodo.5179988).

**Funding:** This research was supported by Basic Science Research Program through the National Research Foundation of Korea (NRF) funded by the Ministry of Education (2020R1A6A1A03044834) and by the Ministry of Science, ICT and Future Planning (2020R1A2C2011081). The funders had no role in study design, data collection and analysis, decision to publish, or preparation of the manuscript.

**Competing interests:** The authors have declared that no competing interests exist.

# Introduction

The recent rapid industrial development of the Korean Peninsula has prompted a significant increase in marine transportation, resulting in increased damage from meteorological disasters (e.g., typhoons, storm surges, floods, and sea fog). The economic losses in aviation, marine and land transportation related to the fog can be comparable to those of winter storms and hurricanes [1]. In particular, fog formation near the sea surface can have a devastating effect on aviation and transport, decreasing horizontal visibility to < 1 km [2]. Simulation studies using numerical meteorological models have been primarily used to investigate the formation and development of sea fog due to the lack of observational data [3–6]. Fu et al. analyzed the effect of the advection process on fog formation through a high-resolution numerical simulation using the Regional Atmospheric Modeling System model [3]. Gao et al. performed numerical simulations of the advection fog in the Yellow Sea using the Fifth-Generation National Center for Atmospheric Research/Penn State Mesoscale Model, and reproduced the primary mechanism of fog formation and movement [4]. Yang et al. investigated the difference in the formation and dissipation of sea fog and its area according to the lowest model-level height of the Weather Research and Forecast (WRF) model [5].

Various uncertainties in numerical models hinder successful fog simulation, and a number of factors, such as initial data [7], spin-up time [8], horizontal and vertical resolution [9, 10], planetary boundary layer (PBL) scheme and microphysics parameterization [11], and sea surface temperature (SST) [12], may affect the results. In particular, SST is an important determinant of ocean–atmosphere exchange, and the differences in spatiotemporal resolution of SST data play an important role in the modeled occurrence and distribution of sea fog [12]. In addition, Cho et al. reported that sea fog occurrences is maximum around the Korean peninsula in July when the difference between air temperature and SST is the highest [13]. Based on a 10 year average (1996 to 1995), the mean frequency of sea fog in July is 5.3 days in the west sea and 2.5 days in the east sea. In fall and winter, this is less than 1 day for most of the seas around the peninsula. The spatio-temporal distributions of sea fog with respect to the three oceans around the Korean Peninsula have been well described in Cho et al [13].

Gultepe et al. provides an overview of a measurement project named C-FOG (Toward Improving Coastal Fog Prediction), which aimed to improve fog prediction [14]. In this review, they introduced a variety of fog studies, e.g. microphysical measurements [1, 15–17], numerical weather predictions and evaluations [1, 6, 15], and visibility parameterizations [2, 18–20]. In the C-FOG project, various instruments for microphysical observations and three-dimensional wind components were used for fog intensity, i.e. visibility. They proposed that liquid water content and droplet number concentration are key parameters for accurate visibility forecasting, and that turbulence kinetic energy dissipation rates are strongly related to the fog life cycle, as well as synoptic weather conditions.

Cold water masses (CWM) are a phenomenon of SST variability, occurring when the coastal water temperatures decrease sharply $\geq 5$ ˚C compared to the surrounding waters. Over the Korean Peninsula, they are created when strong southwesterly winds continue to affect the southeastern coast between June and August. Under these conditions, the surface sea water is swept away to the open ocean, and a CWM is formed by upwelling of the intermediate and deep-sea waters. These events can cause mass die offs in fisheries, or the formation of sea fog [21]. Kim et al. have reported that an average of approximately 24 days of CWM has appeared every year for the past 7 years (2012 to 2018) along the southeastern coast and that the frequency tends to increase in June compared to that in July and August in accordance with the climatic changes such as the change in atmospheric pressure patterns, and the weakening of the monsoon [22].

Numerous studies have aimed to reduce the uncertainty of fog prediction in marine and coastal environments. In particular, to examine the atmospheric effects driving rapidly developing phenomena, such as CWM at various dynamic scales in the ocean, the accuracy of SST data input into the meteorological model is critical; however, most simulations have assumed the short term (< one week) stability of SST. Moreover, the effect of sudden environmental changes (e.g., CWM), on the occurrence and distribution of sea fog in coastal areas has not been studied in detail. Accordingly, this study aimed to analyze the difference in simulated sea fog formation based on SST resolution, and investigated the effects of CWM formation on the occurrence and distribution of sea fog over the Korean Peninsula during the summer of 2016. In addition, the sources of warm and moist airs were also identified with respect to high resolution SST as well as sea fog predictability.

## Methods

### Observation data

In this section, all the observation data are described. First, in-situ station over the ocean were used to analyze SST variation during summer in 2016. These observatories are operated by the Korean Meteorological Agency (KMA), and produces marine meteorological information. The station names are Ulsan (US), Gwangan (GA), Idukseo (ID), Dadaepo (DDP), Oryukdo (ORD), Jangan (JA), and Ganjeolgot (GJG). The geographic locations are shown on the right map in Fig 1. Second, to evaluate the numerical simulations, meteorological measurements inland area were obtained from the Automated Synoptic Observing System (ASOS, Fig 1). Unfortunately, direct sea fog measurements were limited in the study area while inland fog was directly observed by eye at the ground weather stations. Lastly, we applied a fog detection product from the Communication, Ocean, and Meteorological Satellite (COMS) 1, also known as Chollian, which is a geostationary satellite launched in 2010 and ended its own mission in March 2020. Based on the fog detection algorithm [23], it has produced fog area pixels.

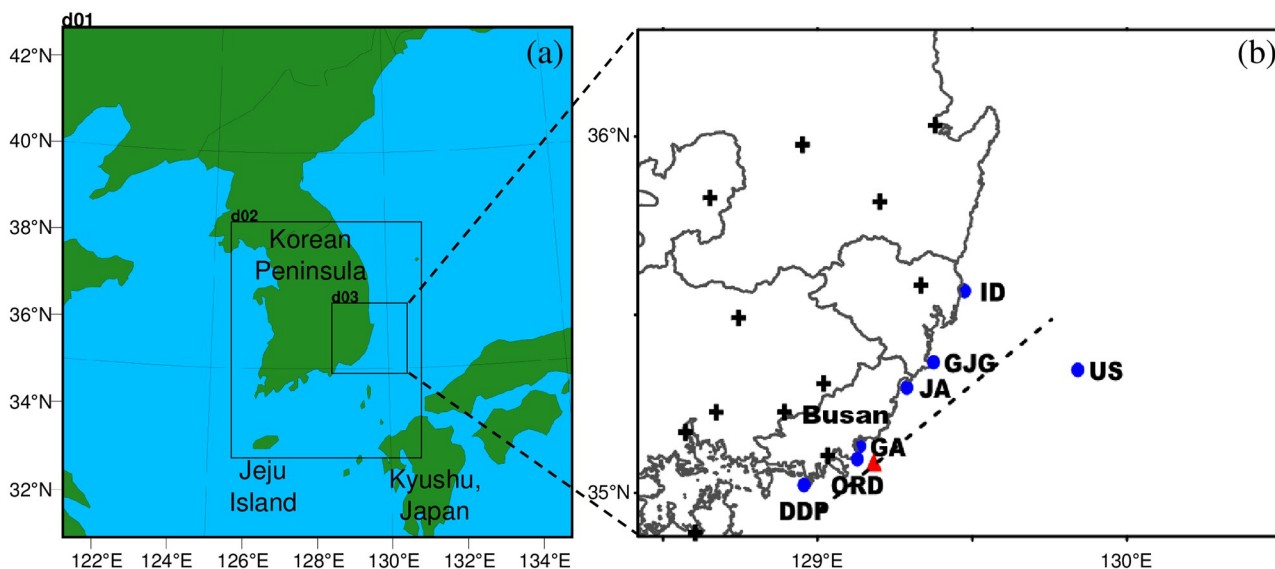

**Fig 1.** (a) Simulation domains for the WRF model used in this study. On the right (b), black plus symbols indicate the ground observatories used for model validation, blue filled circles are the in-situ observations for marine meteorological information operated by the Korean Meteorological Agency, the cross-sectional results were plotted along the dashed line over the ocean, and the red triangle represents the location of vertical analysis. US, Ulsan; GA, Gwangan; ID, Idukseo; DDP, Dadaepo; ORD, Oryukdo; JA, Jangan; and GJG, Ganjeolgot.

## Numerical models

In this study, the numerical model WRF (*v*.3.9.1.1: [24]) was used for the sea fog simulation. The simulation domains were divided into three. To analyze the differences in the sea fog simulation according to the difference in SST resolution in the coastal area, the horizontal resolution was set to 3 km and 1 km for the second (d02) and third (d03) domains, respectively, across the southeastern Korean Peninsula (Fig 1). Detailed experimental configurations of the WRF model are listed in Table 1. Yang et al. studied the significant role of the lowest model level in sea fog simulations in relation to the heat capacity of the air mass [5]. They showed the best performances with the first top layer top at 16 m in terms of the sea fog onset and distributions. If the first layer was lower than 16 m, sea fog developed and dissipated too quickly, and model validation failed over the land area. In this study, the first top layer is approximately 56 m, which is exactly same as the control experiment of Yang et al., and the $\eta$ values are also same below 850 hPa. The simulation period was set as the period from 00 UTC on June 23 to 00 UTC on July 1, 2016, including spin-up. Because actual sea fog was reported by eye measurement at the Busan and Ulsan ASOS stations and through a satellite image on June 30, analyses were conducted around this date. For the initial and boundary condition data of the WRF model, the data from the Final (FNL) Operational Model Global Tropospheric Analyses provided by the National Centers for Environmental Prediction (NCEP) were employed (horizontal resolution, $0.25° \times 0.25°$).

## Real-time global sea surface temperature

The FNL data do not provide separate SST datasets; rather the values corresponding to the skin temperature of the sea were used for the SST variable. To examine the performance of sea fog simulations according to the differences in the resolutions of the SST data, we applied another resolution of the SST data, which were extracted from the skin temperature fields in the FNL data with a horizontal resolution of $1° \times 1°$; therefore, the lower boundary conditions were used for another numerical experiment for a coarser resolution of SST. These two numerical experiments were named as FNL_1° and FNL_1/4°, respectively.

To simulate the sea fog formed by advection of warm and humid air over the cold sea surface, high-resolution SST was required for an accurate consideration of CWM. To use accurate SST data of high resolution in the numerical model, real-time global sea surface temperature (RTG-SST) data from the National Oceanic and Atmospheric Administration (NOAA)/NCEP

**Table 1. Experimental configuration of the WRF simulations.**

|  | d01 | d02 | d03 |
|---|---|---|---|
| **Horizontal grid** | $143 \times 143$ | $148 \times 199$ | $190 \times 178$ |
| **Resolution** | 9 km | 3 km | 1 km |
| **Vertical layers** | 30 $\eta^*$ levels (pressure at top: 50 hPa) | | |
| **Physical process** | Morrison double-moment scheme [25] | | |
|  | RRTMG radiation scheme [26] | | |
|  | MYNN Level 2.5 PBL scheme [27] | | |
|  | Noah Land Surface Model [28] | | |
|  | Kain-Fritsch scheme [29] | | |
| **Initial data** | NCEP FNL Operational Global Analysis data (0.25 degree) | | |

$^*$ $\eta$ levels are defined as $\eta = (p - p_t)/(p_s - p_t)$, where $p$ is pressure, and the subscripts t mean model top and surface, respectively. The full $\eta$ values are 1.000, 0.993, 0.983, 0.970, 0.954, 0.934, 0.909, 0.880, 0.832, 0.784, 0.735, 0.687, 0.604, 0.528, 0.459, 0.398, 0.342, 0.292, 0.247, 0.207, 0.171, 0.139, 0.110, 0.086, 0.065, 0.048, 0.033, 0.020, 0.009, and 0.000 from bottom. First 10 levels below 850 hPa correspond to 0, 59, 143, 254, 393, 569, 793, 1060, 1520 m above ground (or sea) level.

were employed in the simulation (https://www.nco.ncep.noaa.gov/pmb/products/sst/). RTG-SST provides daily SST of equal grid intervals (with the horizontal resolution of 1/12˚) by applying optimal interpolation to satellite-based SST data, in conjunction with buoy and ship observations for the most recent 24 h. Daily real-time test results are provided to ensure the reliability of the data, and in general, the RTG-SST data showed a strong consistency with the moored SST (i.e., test) data due to its ability to properly reflect the rapid SST changes of the Atlantic Gulf Stream and the Pacific Kuroshio Current [30].

In the analyses with FNL_1/4˚ set as the initial and boundary condition data for the SST, the calculations were also performed after replacing the SST data with high-resolution RTG-SST data (RTG_1/12˚ experiment), to ensure the accuracy of the sea fog simulation.

## Results

### Observational evidence and model inputs

To analyze the formation of CWM in the early summer of 2016, the observed daily average SST from the marine meteorological information operated by the Korean Meteorological Agency along the southeast coast, was examined for two months in June and July (Fig 2). According to the seasonal variability of SST, which increases from summer to fall, the SST at the analysis points showed an overall increasing trend; however, on June 24, the SST at the GA, ORD, JA, and GJG sites decreased sharply, recording the minimum SST on July 2nd or 3rd. In contrast, in the US, ID, and DDP sites, SST continued to increase. On June 29 and 30,

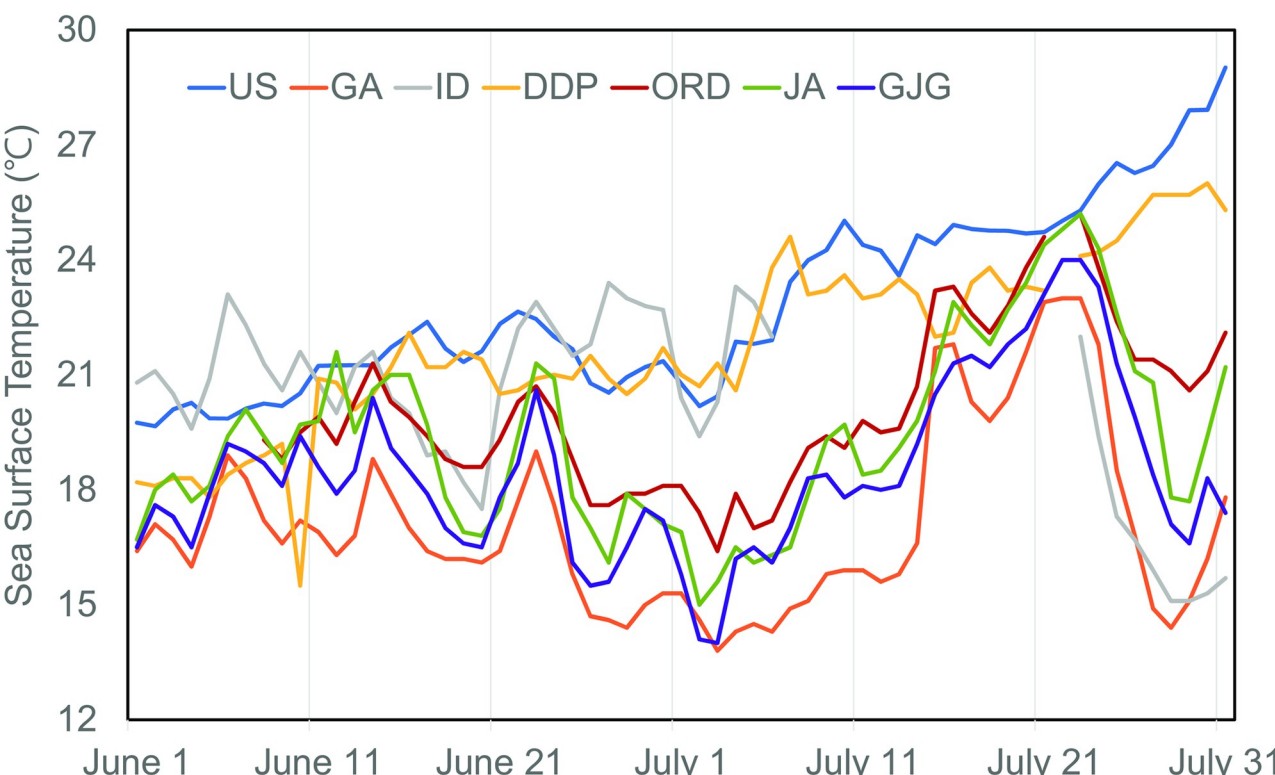

**Fig 2. Daily variations of sea surface temperature (SST) during June–July 2016, collected over three different types of observations: Ocean data buoy (US), light beacon (GA and ID), and coastal wave buoy (DDP, ORD, JA, and GJG) operated by the Korean Meteorological Agency.** Refer to Fig 1 for site name and location.

that is, during actual sea fog occurrence, the grouped mean SSTs were 17.0 ˚C and 21.6 ˚C on June 29, and 16.9 ˚C and 21.9 ˚C on June 30 for the former and latter sites, respectively, (mean differences of 4.7 ˚C and 5.0 ˚C on June 29 and 30, respectively). When comparing GA, the nearest site to Busan city, with DDP, the site furthest south, there was an average difference of 6.9 ˚C in the SST from June 24, the time of initial SST decrease, to July 14, when SSTs sharply increased again. The maximum SST difference reached 9.7 ˚C during this period. These observations showed the formation of a CWM ≥ 5 ˚C colder compared to the surrounding waters. On examining these CWM locations, no CWM formation occurred in the US (open sea), DDP (boundary of the CWM to the south), and ID (boundary of the CWM to the north), enabling a generalized estimation of the spatial distribution of the CWM in the north-south direction, and towards the open sea.

The three SST datasets provided in the numerical model for sea fog simulation in the southeast sea (not the WRF modeled results, but those comparing the lower boundary conditions) were compared to predict how they will affect the atmosphere (Fig 3). The overall distributions

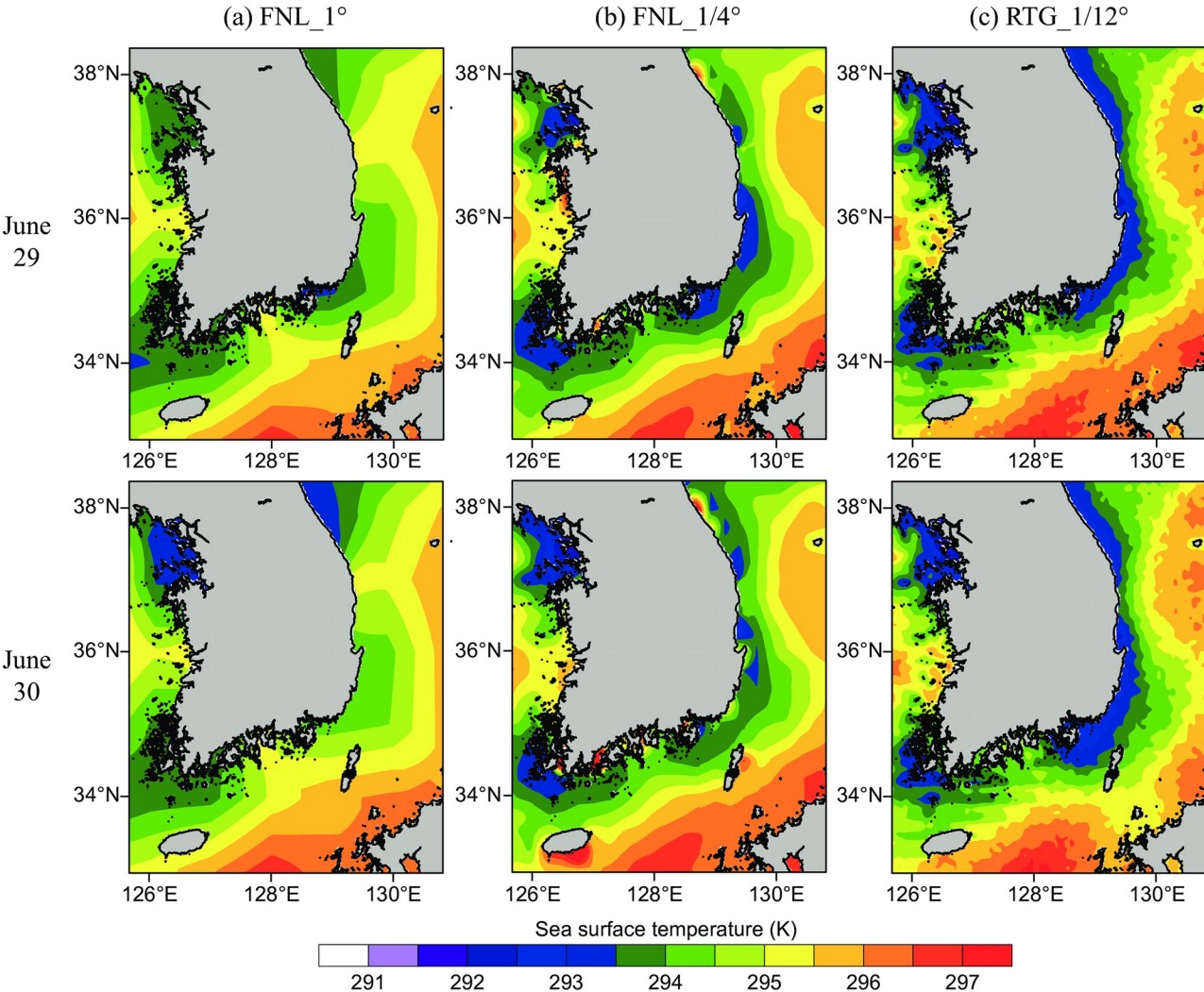

**Fig 3. Horizontal distributions of sea surface temperature (SST) used for (a) FNL_1˚, (b) FNL_1/4˚, and (c) RTG_1/12˚ simulations on June 29 (top row) and June 30 (bottom row).**

of SST along the southeast coast of the Korean Peninsula appeared similar across all datasets, with relatively low temperatures along the shoreline increasing towards the open sea; however, remarkable differences were observed in the local SST distribution, depending on the dataset type and resolution. On June 29, in the sea adjacent to the southeast coastline, the FNL_1˚ dataset had an SST of 294 K, the FNL_1/4˚ reported 293 K, and 293 K was more widely distributed along the shoreline in RTG-SST. On June 30, the RTG_1/12˚ dataset showed a broad distribution of lower SSTs, reflecting the CWM. Additionally, when the data was input by interpolation at the horizontal resolution of the simulated model, the resolution differences inferred the existence of varied boundary conditions within the lower layer of atmosphere, and its consequent cooling effect.

Although the SST outputs of the FNL and RTG datasets were generated based on buoy and satellite data, these data were compared to in-situ observations adjacent to the coastal region for June 29 and 30 (Table 2). The SST values at the GA, ORD, JA, and GJG sites were higher than the observed values for all three types of datasets; however, the RTG datasets were lower than either of the FNL datasets. The mean RMSE for all points over the simulation period from June 23 to 30 were 4.4 ˚C, 4.2 ˚C, and 2.7 ˚C for FNL_1˚, FNL_1/4˚, and RTG_1/12˚, respectively, indicating that error was reduced with increasing SST resolution.

Conventionally, it is assumed that there is no change in SST when the meteorological numerical simulation is performed for less than 1 week, as sea dynamics are slower than atmosphere dynamics. Accordingly, the initial SST distribution is used throughout the simulation period; however, small-scale, rapidly developing oceanic phenomena lasting for several days, as in the case of CWM, act as an important driving factor between sea and the lower atmosphere. The high-resolution SST data must, therefore, be provided in the numerical models whenever possible.

Observational evidence of the sea fog in this period was identified in the COMS1 product. An example of produced sea fog signal from COMS 1 is shown in Fig 4. Orange pixels indicate the fog detected area through a threshold test and time continuity test, and this is a relatively reliable result with less possibility of a pollutant pixel suddenly appearing from a cloud or the ground surface. The sea fog was clearly observed in the southeastern sea of the Korean Peninsula in the night-time on June 30, 2016.

## Model validation

To evaluate the WRF model performance, the simulation results from June 25 to 30 were compared with 2 m temperature and 10 m wind speed data collected across 12 meteorological

**Table 2. Sea surface temperature (SST) observed in in-situ stations (Obs.) and used in the three simulations (FNL_1˚, FNL_1/4˚, and RTG_1/12˚) on June 29 and 30, 2016.**

| DATE | Case | Location | | | | | | |
|---|---|---|---|---|---|---|---|---|
| | | US | GA | ID | DDP | ORD | JA | GJG |
| June 29 | Obs. | 21.2 | 15.0 | 22.8 | 20.9 | 17.9 | 17.5 | 17.5 |
| | FNL_1˚ | 21.1 | 22.7 | 20.9 | 20.2 | 20.4 | 20.5 | 20.7 |
| | FNL_1/4˚ | 20.7 | 21.9 | 20.2 | 20.1 | 20.4 | 20.6 | 20.4 |
| | RTG_1/12˚ | 20.8 | 19.9 | 19.9 | 19.8 | 19.9 | 19.9 | 19.9 |
| June 30 | Obs. | 21.4 | 15.3 | 22.7 | 21.7 | 18.1 | 17.1 | 17.2 |
| | FNL_1˚ | 21.3 | 26.9 | 21.2 | 21.2 | 21.2 | 21.2 | 21.2 |
| | FNL_1/4˚ | 20.7 | 26.1 | 20.6 | 20.9 | 21.1 | 22.0 | 21.1 |
| | RTG_1/12˚ | 20.7 | 19.7 | 20.0 | 19.5 | 19.6 | 19.8 | 19.8 |

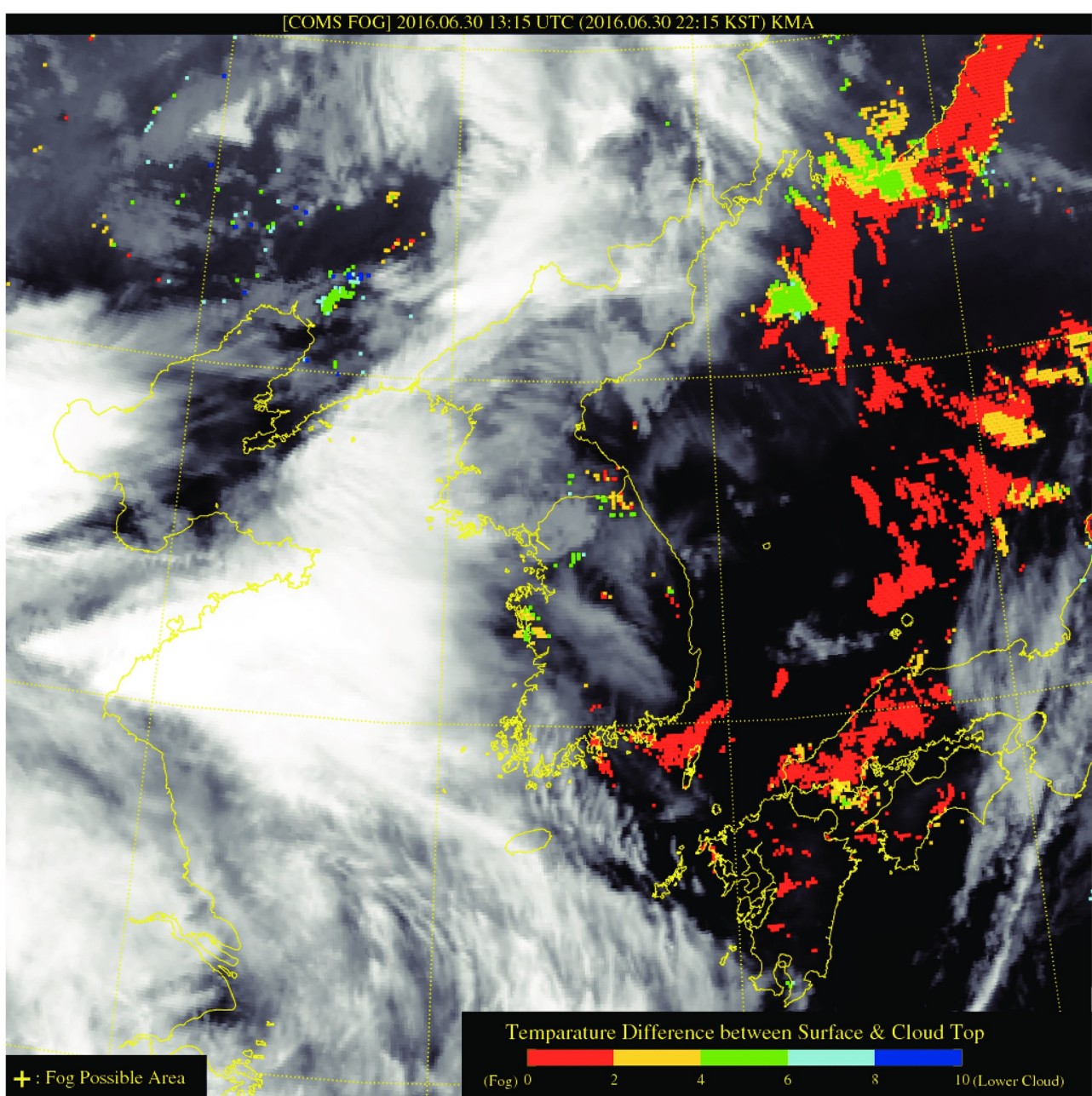

**Fig 4. Communication, Ocean, and Meteorological Satellite (COMS) 1, also known as Chollian image of the sea fog episode on June 30, 2016.** Orange color code indicates the possible area of sea fog.

observation points in the d03 area (plus sign in Fig 1). Except for the SST input data, the initial and boundary conditions were identical for all experiments, and the effects of SST changes on the inland ground observation points was insignificant. Accordingly, model validation was performed on the FNL_1/4° experiment, and the results of the statistical analyses are presented in Table 3. The temperature results obtained from the WRF model showed good agreement with the observations (RMSE = 1.83°C, MBE = 0.22°C, IOA = 0.94, and R = 0.88). The wind

**Table 3. Statistical indices of FNL_1/4 simulation evaluating the 2 m temperature (T2) and 10 m wind speed (WS) at 12 observation sites in the innermost domain (d03).**

| [a]Index | T2 (˚C) | WS (m·s⁻¹) |
|---|---|---|
| MEAN | 22.87 | 2.79 |
| RMSE | 1.83 | 1.91 |
| MBE | 0.22 | 1.04 |
| IOA | 0.94 | 0.64 |
| R | 0.88 | 0.55 |

[a]MEAN: $(1/N) \sum_{i=1}^{N} O_i$, RMSE: $\sqrt{(1/N) \sum_{i=1}^{N} (M_i - O_i)^2}$, MBE: $(1/N) \sum_{i=1}^{N} (M_i - O_i)$, IOA: $1 - \frac{\sum_{i=1}^{N} (M_i - O_i)^2}{\sum_{i=1}^{N} (|M_i - \bar{O}| + O_i - \bar{O})^2}$,

and R: $\frac{\sum_{i=1}^{N} (M_i - \bar{M})(O_i - \bar{O})}{\sqrt{\sum_{i=1}^{N} (M_i - \bar{M})^2} \sqrt{\sum_{i=1}^{N} (O_i - \bar{O})^2}}$, where $M$ and $O$ are model and observation, respectively.

speed results tended to overestimate the observed values (RMSE = 1.91 m·s⁻¹, MBE = 1.04 m·s⁻¹), although a significant relationship was verified (IOA = 0.64, and R = 0.55). Although the WRF simulation results at each resolution were insignificant with respect to the SST data for the validation points, sea fog formation along coastal areas evidently differed with complex shorelines and in the sea.

## Horizontal distributions of upward moisture flux over the ocean

Fig 5 shows the vertical moisture flux (VMF) on the sea surface at 1200 LST on June 29 and 30. It can be seen that strong southwest winds blew continuously off the coast of Busan on both dates. In the lower layer, these winds create favorable conditions for inducing upwelling. The VMF on June 29 indicated that there was a strong supply of water vapor (> 0.018 kg·m⁻²·s⁻¹) to the atmosphere from the seas east of Jeju Island, and west of Kyushu, Japan. Due to the persistence of warm pools in the same area acting as strong sources of water vapor (Fig 3), this varied depending on the air temperature. The water vapor supply appeared the strongest in the RTG_1/12˚ experiment, whereas the horizontal size of the warm pool was approximately half that of the other datasets in FNL_1˚. In the coastal VMF adjacent to Busan, the value of FNL_1˚ was the highest among the datasets, indicating that the cold water mass effect was invisible at this scale. When examining RTG_1/12˚ at the same location, VMF was near zero, implying that the sea fog results described in Section 3.4 were not attributable to local sources of water vapor. The continuous supply of water vapor in the South Sea of the Korean Peninsula is thus transported from the open sea to offshore Busan via the dominant southwest winds over the study period, showing that condensation caused by the cooling of the lower layer can result in sea fog formation.

## Verification of simulated sea fog

Fig 6 shows the results of the simulated sea fog event that occurred on June 30, 2016. Through analysis of the horizontal distribution of clouds in the lowest model layer, the formation of sea fog was determined. At 0000 LST, all three analyses simulated clouds in the lowest layer of the southwest coast near Jeju Island. This was not sea fog confined to the lower layer of the atmosphere, but a synoptic-scale low cloud, stretching up to an altitude of approximately 500 m or higher (the upper layer image is omitted). Conversely, the lowest layer cloud over the southeastern sea was only captured in the RTG_1/12˚ simulation. There were no clouds upwards from the second layer; thus, the clouds present in the RGT_1/12˚ simulation can be regarded

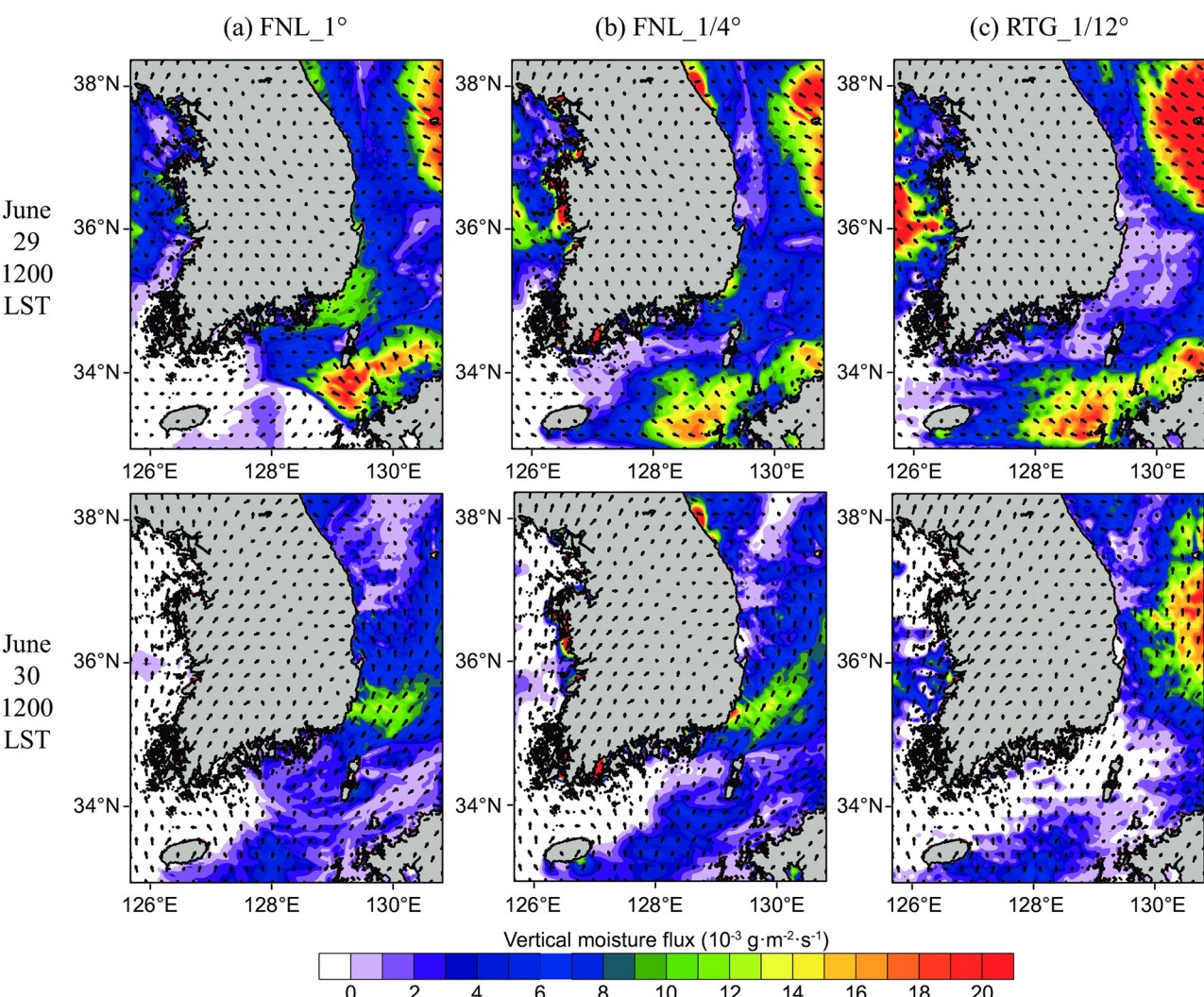

**Fig 5. Horizontal distributions of Vertical Moisture Flux (VMF) from the ground and ocean surface for: (a) FNL_1, (b) FNL_1/4˚, and (c) RTG_1/12˚ simulations on June 29 (top row) and 30 (bottom row).**

as low sea fog. The FNL_1/4˚ simulation showed a local distribution of sea fog, but its formation was unclear. This indicates that not only the inclusion of the CWM in the SST data input is an important influence, but also the distribution and persistence of these lower SSTs over a wide area are vital. There was a continuous southwest wind along the southeast coast on June 30, and sea fog formed and dissipated repeatedly until nighttime (lower panels of Fig 6). In particular, the spatial coverage of the simulated sea fog at 2200 LST on June 30 (Fig 6c) agreed well with the satellite product (Fig 4) using high resolution SST data (the satellite data at 0000 LST on June 29 is not shown due to lower cloud contaminant).

## Integrated water vapor transport (IVT)

Fig 5 corresponds to the vertical water vapor supply from the ocean to the atmosphere, whereas Fig 7 shows the results of the horizontal transport of water vapor in the lower atmosphere. The integrated water vapor transport (IVT) was analyzed to examine whether the sea

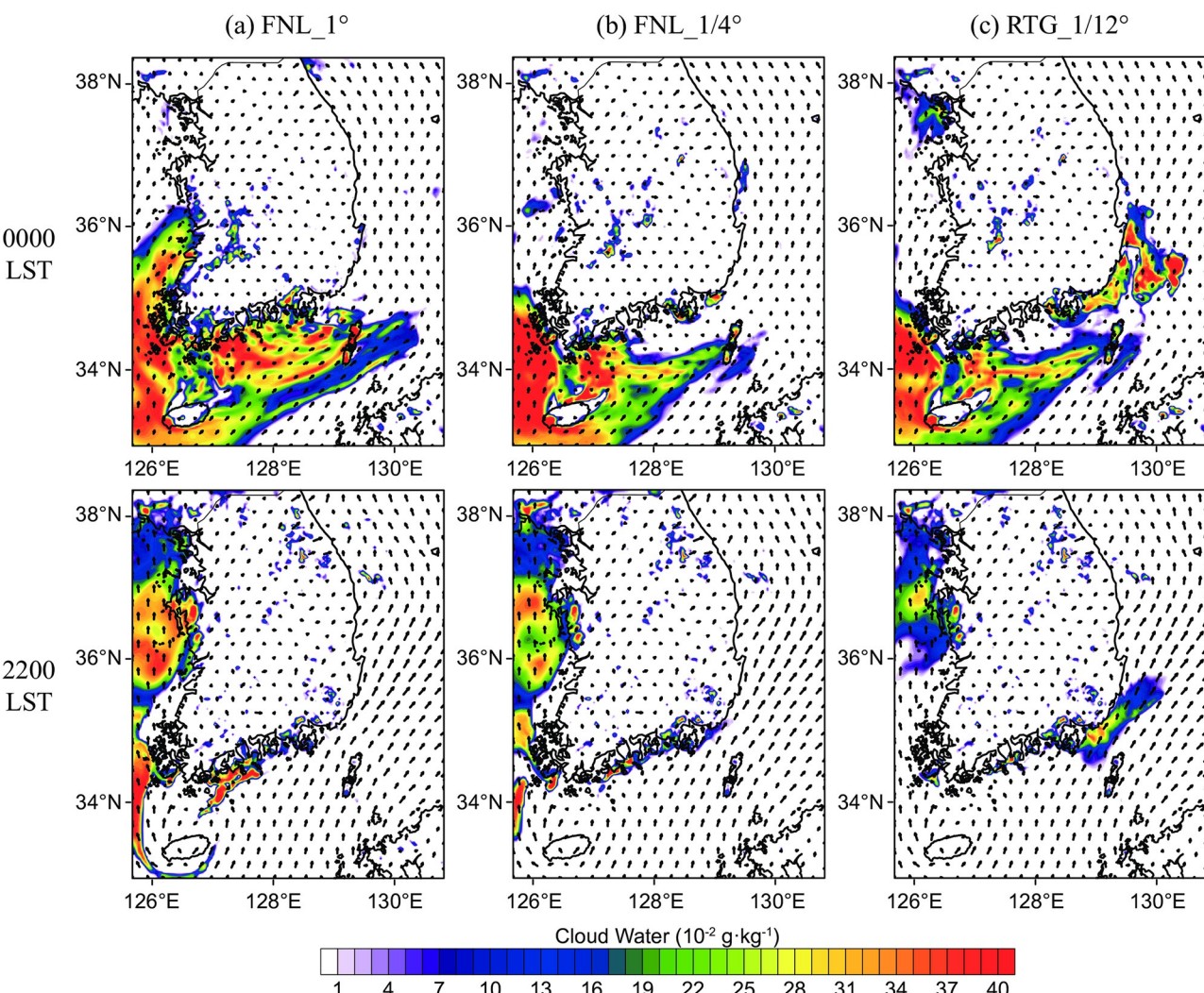

**Fig 6. Spatial distributions of cloud water in the 1st model layer for: (a) FNL_1˚, (b) FNL_1/4˚, and (c) RTG_1/12˚ simulations at 0000 LST (top row) and 2200 LST (bottom row) on June 30, 2016.**

fog off the coast of Busan (Fig 6) was formed by water vapor transported from the south. For a detailed calculation, Eq. (7) of Banacos and Schultz [31] was used, and if divergence is excluded, it represents IVT as follows:

$$IVT = \rho \int_{sfc}^{250m} q\vec{V}_h dz = \frac{1}{g} \int_{P_{sfc}}^{P_{250m}} q\vec{V}_h dp$$

where after calculating the flux by multiplying the horizontal wind component ($\vec{V}_h$) by the water vapor value ($q$), the mass-corrected weighting ($dp/\rho$) was multiplied by the pressure layer thickness for each layer. To obtain the water vapor transport in the lower boundary layer, integration was performed from the third layer to the (sea) surface, which can be interpreted as the average water vapor transport up to a height of approximately 250 m. The unit of IVT is kg·m$^{-1}$·s$^{-1}$, or the number of kilograms of water vapor that move across a distance of 1 m in 1 s, according to the average wind in the specified layer for integration. Fig 7 shows the IVT at

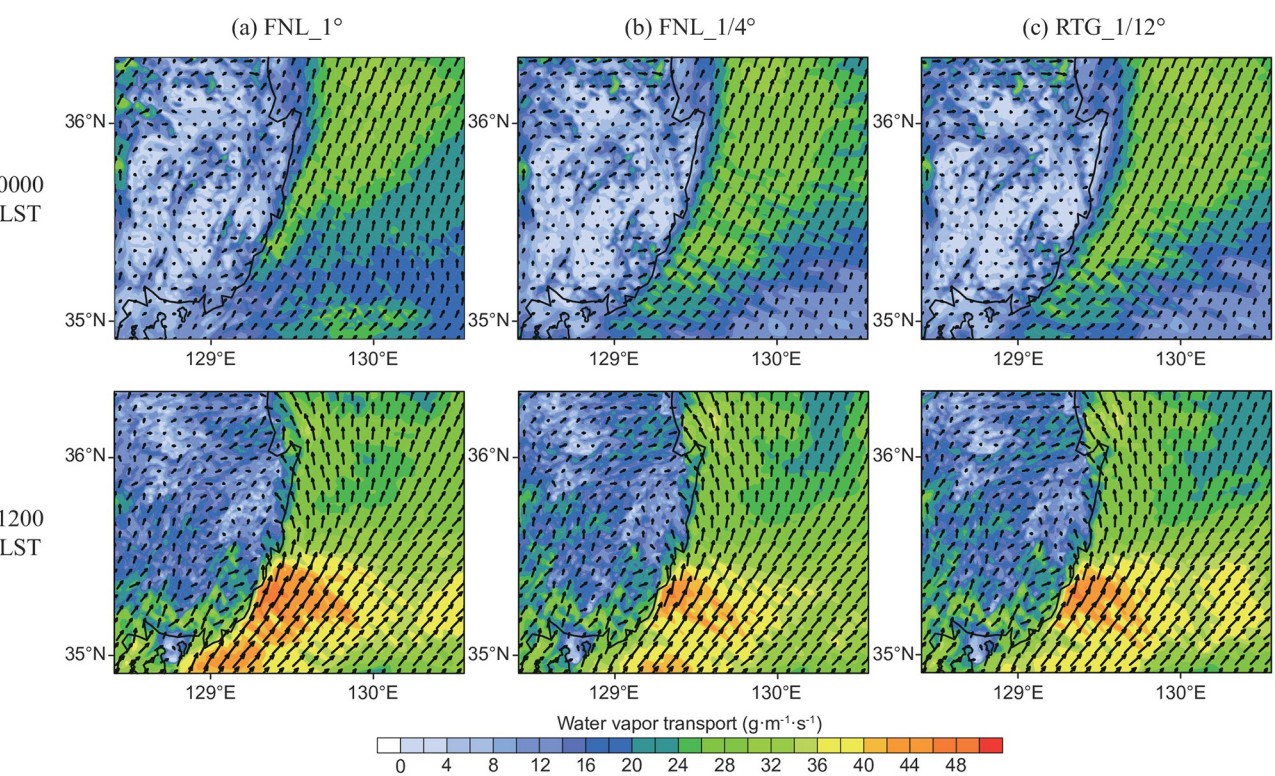

**Fig 7. Vertically integrated water vapor transport (IVT) for: (a) FNL_1˚, (b) FNL_1/4˚, and (c) RTG_1/12˚ simulations at 0000 LST (top row) and 1200 LST (bottom row) June 30, 2016.**

0000 LST and 1200 LST on June 30 for Domain 3, and the magnitude of the vector is represented by a contour. In all three analyses, strong water vapor transport from the southern border was apparent. For the FNL_1/4˚ and RTG_1/12˚ datasets, water vapor was transported from the warm pool source area between Jeju Island and Kyushu (Fig 5), to the southeastern coast of the Korean Peninsula (IVT ≈ 30 kg·m$^{-1}$·s$^{-1}$ at 0000 LST, June 30). For FNL_1˚, the warm pool area was the smallest among the datasets; however, due to the local supply of water vapor in the southeast sea, stronger water vapor transport was observed at 1200 LST on June 30. It should be noted that although the IVT results were similar between the FNL_1/4˚ and RTG_1/12˚ experiments, sea fog formation was only confirmed in the RTG_1/12˚ experiment (Fig 6); thus, even under the conditions of similar total water vapor transport amounts, the occurrence of sea fog can only be properly simulated when considering the precise CWM.

## Vertical extent of simulated sea fog

Vertical analyses were conducted for a three-dimensional investigation of the mechanisms of sea fog formation and structure on June 30, 2016. For the most detailed analysis, only the RTG_1/12˚ simulation, in which the sea fog was most clearly shown in the d03 area, was examined. Along the southeast coast, sea fog showed some variation in vertical thickness and horizontal dimension, but remained throughout the day. Fig 8 shows the process of sea fog development at 2100 LST (Fig 8d) where the atmosphere was relatively stable, and the vertical cross section of the hourly cloud water mixing ratio is displayed as the dashed line depicted in Fig 1. Because the wind speed in the vertical direction (*w*) is an order of magnitude weaker than the horizontal wind speed, it was increased 10-fold to display the two vectors

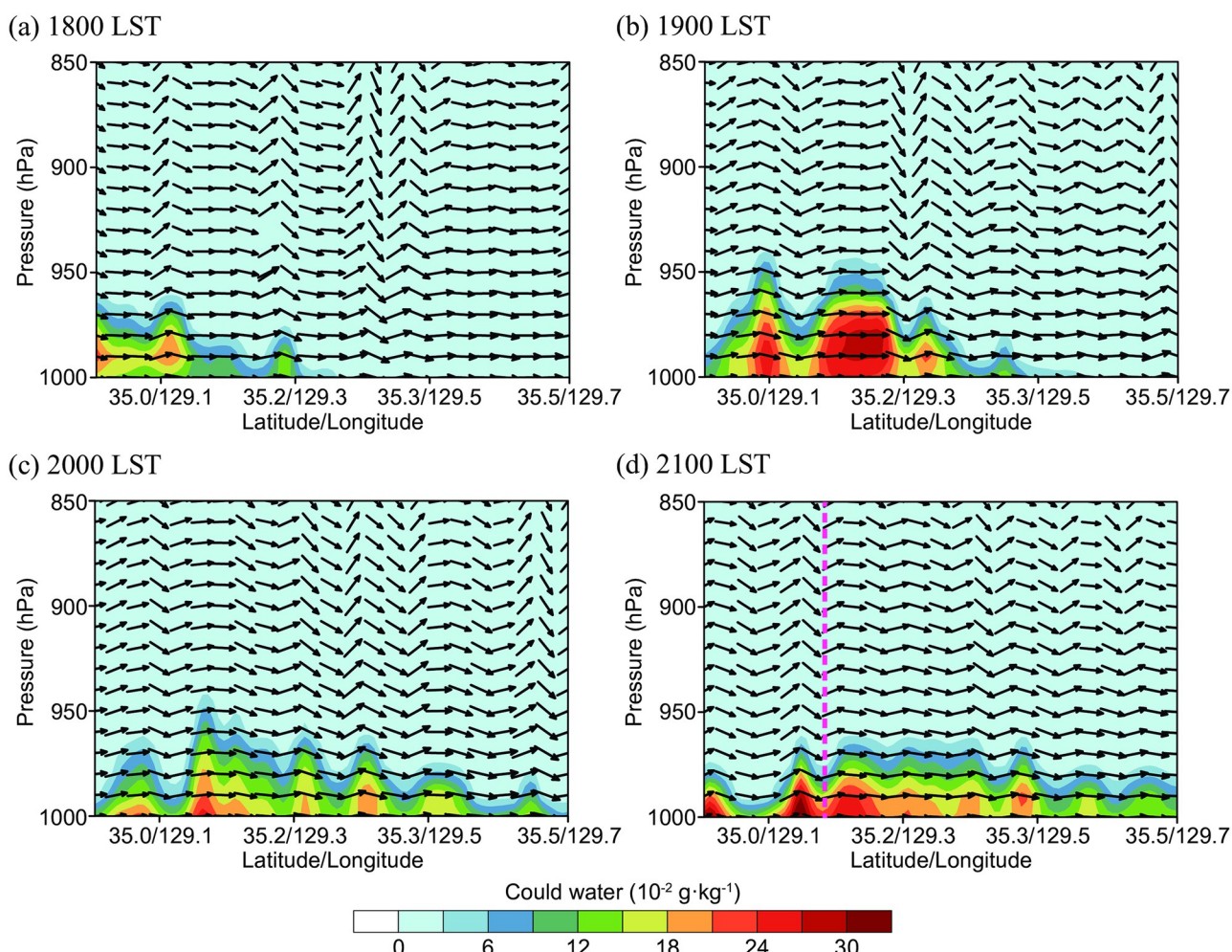

(a) 1800 LST
(b) 1900 LST
(c) 2000 LST
(d) 2100 LST

Could water ($10^{-2}$ g·kg$^{-1}$)

0    6    12    18    24    30

**Fig 8. Time variations of cloud water on the cross-sectional plane along the dashed line in Fig 1 for the RTG_1/12˚ simulation on June 30, 2016.**
The pink dashed line on the bottom right panel (d) represents the location of the vertical distributions of cloud and vapor water mixing ratio in Fig 9.

simultaneously. The vector pointing to the right in the figure indicates the southwesterly wind, which was the sustained prevailing wind at the boundary layer scale. Accordingly, the water vapor introduced at 1800 LST condensed near the sea surface, resulting in fog, the extent of which gradually expanded to the northeast.

To examine whether sea fog was formed by the condensation of water vapor transported from the southwest through the southern boundary of the d03 domain over a CWM, the vertical distributions of water vapor and cloud water at a fixed location were analyzed for the three experiments. The analysis locations are represented in Fig 1 with a red triangle. The vertical distribution can be seen in Fig 9, along the dashed pink line marked in the cross-section of Fig 8 at 2100 LST on June 30. First, the distribution of water vapor generally decreased from the lower to upper layers for all three experiments; however, in Fig 9a, only RTG_1/12˚ showed lower water vapor values below approximately 990 hPa (corresponding to a height of approximately 140 m above sea level). The difference in water vapor at approximately 1 g·kg$^{-1}$ can be confirmed by the presence of condensed clouds (Fig 9b), which were only present in the lowest layer of the RTG_1/12˚ experiment. The weak cloud formation from approximately 2–300 m

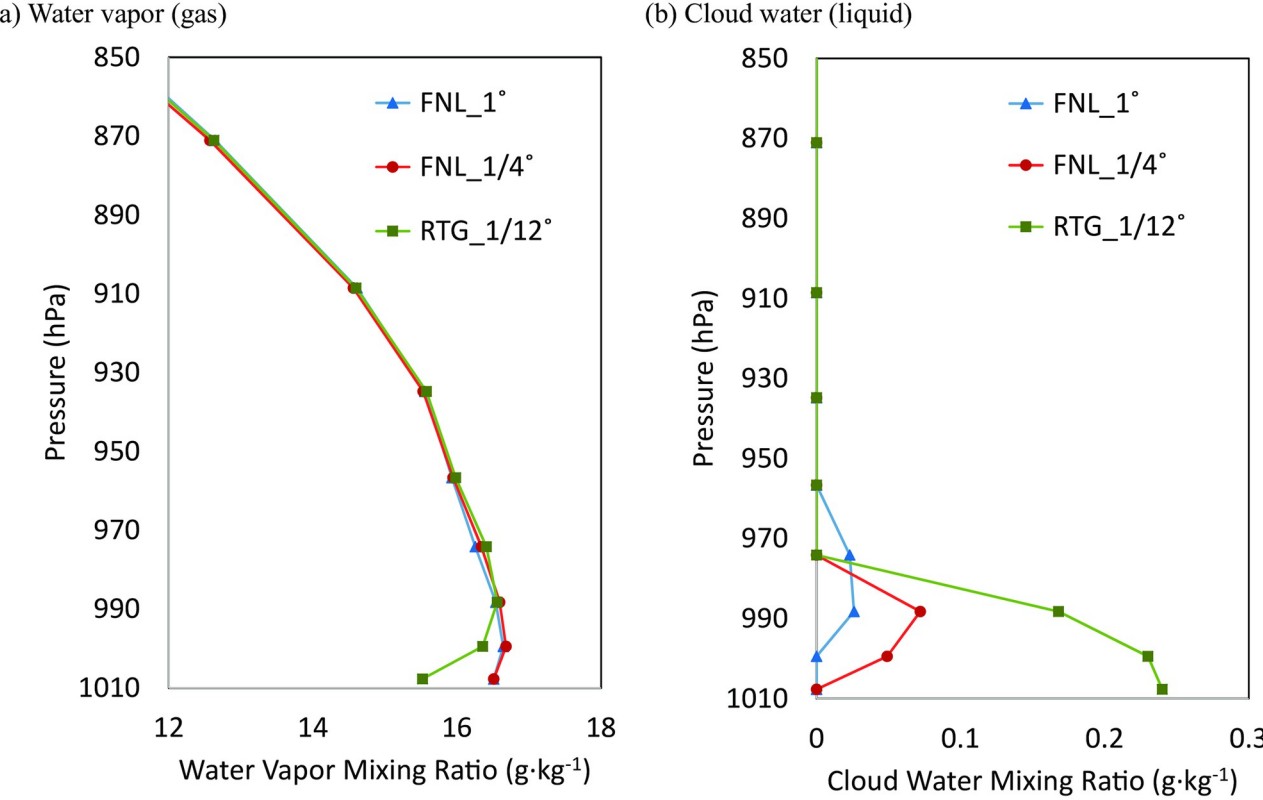

**Fig 9. Vertical distributions of: (a) water vapor, and (b) cloud water mixing ratios for FNL_1˚ (blue), FNL_1/4˚ (red), and RTG_1/12˚ (green) simulations at 2100 LST, June 30, at the fixed location marked with a triangle in Fig 1.**

above sea level can be regarded as the cloud moving from nearby areas, and not the cloud formed by local condensation. The water vapor decreased in the 1st and 2nd layers from the sea surface for the RTG_1/12˚ experiment, whereas the cloud water was present in all the 1st, 2nd, and 3rd layers. This was attributed to the sea fog formation by the lower atmospheric turbulence developed vertically.

To identify the condensation process due to radiative cooling at the surface, the differences between dew point temperature ($T_d$) and SST were analyzed (Table 4). As Cho et al. suggested, we used $T_d$ that includes the meaning of both humidity and air temperature. Table 4 gives the

**Table 4. Mean results of the difference between dew point temperature and sea surface temperature ($T_d$-SST) where the in-situ stations are located for the three simulation cases on June 29 and 30, 2016.** In the parentheses, daily mean $T_d$s are also shown. The values in a bold character mean the potential for radiative cooling at the surface.

| DATE | Case | Location | | | | | | |
|---|---|---|---|---|---|---|---|---|
| | | US | GA | ID | DDP | ORD | JA | GJG |
| **June 29** | FNL_1˚ | -0.84 (20.3) | -3.01 (19.7) | -1.07 (19.8) | -0.21 (20.0) | -0.50 (19.9) | -0.61 (19.9) | -0.70 (20.0) |
| | FNL_1/4˚ | -0.66 (20.0) | -2.32 (19.6) | -0.41 (19.8) | -0.22 (19.9) | -0.60 (19.8) | -0.83 (19.8) | -0.48 (19.9) |
| | RTG_1/12˚ | -0.62 (20.2) | -0.74 (19.2) | -0.50 (19.4) | -0.72 (19.1) | -0.62 (19.3) | -0.57 (19.3) | -0.49 (19.4) |
| **June 30** | FNL_1˚ | -0.19 (21.1) | -6.03 (20.9) | -0.42 (20.8) | -0.23 (21.0) | -0.29 (20.9) | -0.34 (20.9) | -0.26 (20.9) |
| | FNL_1/4˚ | **0.23** (20.9) | -5.60 (20.5) | -0.05 (20.6) | -0.34 (20.6) | -0.54 (20.6) | -1.30 (20.7) | -0.38 (20.7) |
| | RTG_1/12˚ | -0.45 (20.2) | **0.45** (20.1) | -0.18 (19.8) | **0.14** (19.6) | **0.23** (19.8) | -0.01 (19.8) | **0.15** (19.9) |

daily mean $T_d$-SST on June 29 and 30, 2016 for each simulation case at the representative locations. In addition, daily mean $T_d$s are shown in parentheses below. If the Td-SST is positive, there is potential for radiative cooling at the surface, thus leading to condensation. On June 30, the RTG_1/12˚ case showed positive values at GA, DDP, ORD, and GJG. Spatial means resulted in -1.11 K, -1.14 K, and +0.04 K, for FNL_1˚, FNL_1/4˚, and RTG_1/12˚, respectively on June 30, 2016. Although $T_d$ for the RTG_1/12˚ case was lower on average than those for the FNL_1˚ and FNL_1/4˚ experiments due to the sea fog formation, the positive Td-SST attests strong radiative cooling by cold SST. Our results from RTG_1/12˚ is consistent with Cho et al. based on observed data, which reported the highest frequency of sea fog when $T_d$-SST is 0.0–1.9 K in the southern sea of the Korean Peninsula [13].

## Discussion and conclusions

In this study, numerical simulations were conducted for the prediction and analysis of sea fog that frequently occurs along the southeastern coast of the Korean Peninsula in the early summer due to the difference in heat capacity between the atmosphere and the ocean. Most oceanic motion takes place over long periods, with wide areas of distribution, but CWM develop rapidly over relatively narrow areas, and disappear after a few days. To accurately consider the effect of CWM, SST data of different resolutions were input into the WRF numerical weather model.

Based on the reference resolution of 0.25˚ FNL, in which the skin temperature of the global reanalysis field was employed as SST data (FNL_1/4˚), FNL 1˚ SST data (FNL_1˚ experiment) for a coarse resolution, and 1/12˚ RTG-SST data (RTG_1/12˚ experiment) for a high resolution were used to assess the sea fog simulation characteristics according to the resolution. The results indicated that sea fog formation from the effects of CWM were simulated only in the RTG_1/12˚ experiment using high-resolution data. In addition to the presence or absence of CWM, RTG_1/12˚ also maintained the widest horizontal distribution extent of CWM, and thus is suitable for simulating sea fog formed by condensation in the lower layer of the atmosphere from large air-sea temperature differences. This well-presented CWM in the RTG_1/12˚ led to a good agreement of horizontal distributions with the satellite image. In addition, it was identified that the difference in resolution of the input SST data caused a difference in the size and strength of the warm pool that provides water vapor from the open sea region of the South Sea of the Korean Peninsula to the atmosphere, also affecting the amount and direction of water vapor transported to the southeast coast. As a result of examining the horizontal water vapor transport and the vertical structure of the generated sea fog using a high-resolution meteorological simulation (1 km grid size), the water vapor continuously introduced by the southwesterly winds from June 29 to 30 created a fog event at dawn and in the evening of June 30, which developed from 30 m to 450 m in height. Furthermore, through the concurrent analysis of the vertical distribution of water vapor and cloud water, it was demonstrated that the sea fog developed along the southeast coast was not formed locally, but instead by the condensation of transported water vapor encountering a CWM.

Even the input SST for the RTG_1/12˚ experiment overestimated the observed SST at the selected sites, e.g., the mean bias in GA, ORD, JA, and GJG was +2.9 K (Table 2), thus the sea fogs were successively simulated. Therefore, we conducted additional sensitivity tests of SST conditions over the sea close to the southeastern coastline. In these tests, the SSTs were increased by 1 K and 2 K, and decreased by 1 K where the original SST showed less than 239.5 K on June 29 and 30, which means warming and cooling the CWM (S1 Fig). We confirmed that the CWM area with a 1 K lower SST intensified sea fog and its distribution while that with 1 K higher SST showed the limited sea fog area (S2 Fig). In addition, it was unable to capture

any sea fog area when the CWM was increased by 2 K with the same warm pool sources in the open sea. This sensitivity test implies that a sea fog event can be predicted depending on how cold a CWM is even though the observation showed colder SST.

As reported in Cho et al., the CWM and sea fog in the southeastern ocean are strongly related and therefore the sea fog in this area occurred frequently in the summer season [13]. Previously, we have acquired quite similar conclusions for the sea fog event in 2011 that the high-resolution SST is crucial for predicting sea fog with the effect of CWMs (S3 Fig). The SST data, however, was the Operational Sea Surface Temperature and Sea Ice Analysis (OSTIA), and the detail configurations were different. Nevertheless, the previous simulations convinced that the sea fog is attributed to the CWM along the southeastern coast of the Korean Peninsula. Although there were no quantitative measurements of sea fog observations available, sea fog was examined through anecdotal fog reports via visual observation and photographic data as well as a satellite algorithm. In the future, it will be necessary to perform more detailed analyses on the importance of lower boundary condition data of numerical meteorological models by combining them with visibility observations from the sea or satellite data. Real-time calculations of CWM over the ocean would also be possible to couple regional-scale ocean models (e.g., Regional Ocean Modeling System—ROMS) with the WRF model [32]. It is expected that the findings of this study will contribute to the reduction of ship accidents via the accurate numerical simulation of sea fog due to the low visibility that frequently occurs on the southeastern coast of the Korean Peninsula in the early summer.

## Supporting information

**S1 Fig. Horizontal distributions of sea surface temperature (SST) used for the sensitivity tests: (a) RTG_1/12˚-1K, (b) RTG_1/12˚, (c) RTG_1/12˚+1K, and (d) RTG_1/12˚+2K simulations on June 29 (top row) and June 30 (bottom row).** The rectangular with the dashed line indicates the SST adjusted area in which SST was decreased by 1 K or increased by 1 and 2 K if it is lower than 293.5 K.
(PDF)

**S2 Fig. Spatial distributions of cloud water in the 1st model layer for the sensitivity tests: (a) RTG_1/12˚-1K, (b) RTG_1/12˚, (c) RTG_1/12˚+1K, and (d) RTG_1/12˚+2K simulations at 0000 LST (top row), 0400 LST (middle row) and 2200 LST (bottom row) on June 30, 2016.** The rectangular with the dashed line indicates the SST adjusted area the same as in S1 Fig.
(PDF)

**S3 Fig. Spatial distributions of cloud water in the 1st model layer for another sea fog event related to cold water mass over ocean: (a) the simulation using Operational Sea Surface Temperature and Sea Ice Analysis (OSTIA) with 1/20˚ horizontal resolution, and (b) the simulation using skin temperature fields in the FNL with 1˚ horizontal resolution at 2100 LST on July 11 (left), 0000 LST on July 12 (center), and 0300 LST (right) on June 12, 2011.**
(PNG)

## Author Contributions

**Conceptualization:** Soon-Young Park, Soon-Hwan Lee.

**Formal analysis:** Soon-Young Park.

**Investigation:** Jung-Woo Yoo.

**Methodology:** Soon-Young Park.

**Project administration:** Cheol-Hee Kim.

**Supervision:** Soon-Hwan Lee.

**Validation:** Soon-Young Park, Sang-Keun Song.

**Visualization:** Jung-Woo Yoo.

**Writing – original draft:** Jung-Woo Yoo.

**Writing – review & editing:** Soon-Young Park, Sang-Keun Song, Cheol-Hee Kim.

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
