## [Decision Letter · Decision Letter 0]

8 Oct 2021

PONE-D-21-26199Numerical study on advective fog formation and its characteristic associated with cold water upwellingPLOS ONE

Dear Dr. Lee,

Thank you for submitting your manuscript to PLOS ONE. After careful consideration, we feel that it has merit but does not fully meet PLOS ONE’s publication criteria as it currently stands. Therefore, we invite you to submit a revised version of the manuscript that addresses the points raised during the review process. In particular, you will see that one reviewer asks for several formal improvements in the presentation aspects of the article, while the other reviewer questions the novelty of the work and impact of the SST data used therein. You therefore need to carefully address these points in the revised version of the manuscript.

We look forward to receiving your revised manuscript.

Kind regards,

Andrea Storto

Academic Editor

PLOS ONE

Journal Requirements:

"This research was supported by Basic Science Research Program through the National Research Foundation of Korea (NRF) funded by the Ministry of Education (2020R1A6A1A03044834) and by the Ministry of Science, ICT and Future Planning (2020R1A2C2011081)"

"This research was supported by Basic Science Research Program through the National Research Foundation of Korea (NRF) funded by the Ministry of Education (2020R1A6A1A03044834) and by the Ministry of Science, ICT and Future Planning (2020R1A2C2011081). The funders had no role in study design, data collection and analysis, decision to publish, or preparation of the manuscript"

3. We note that Figure 1 in your submission contain copyrighted images. All PLOS content is published under the Creative Commons Attribution License (CC BY 4.0), which means that the manuscript, images, and Supporting Information files will be freely available online, and any third party is permitted to access, download, copy, distribute, and use these materials in any way, even commercially, with proper attribution. For more information, see our copyright guidelines: http://journals.plos.org/plosone/s/licenses-and-copyright.

2. If you are unable to obtain permission from the original copyright holder to publish these figures under the CC BY 4.0 license or if the copyright holder’s requirements are incompatible with the CC BY 4.0 license, please either i) remove the figure or ii) supply a replacement figure that complies with the CC BY 4.0 license. Please check copyright information on all replacement figures and update the figure caption with source information. If applicable, please specify in the figure caption text when a figure is similar but not identical to the original image and is therefore for illustrative purposes only

4. We note that you have stated that you will provide repository information for your data at acceptance. Should your manuscript be accepted for publication, we will hold it until you provide the relevant accession numbers or DOIs necessary to access your data. If you wish to make changes to your Data Availability statement, please describe these changes in your cover letter and we will update your Data Availability statement to reflect the information you provide

Reviewers' comments:

Reviewer's Responses to Questions

**Comments to the Author**

1. Is the manuscript technically sound, and do the data support the conclusions?

Reviewer #1: Partly

Reviewer #2: Partly

2. Has the statistical analysis been performed appropriately and rigorously? 

Reviewer #1: No

Reviewer #2: No

3. Have the authors made all data underlying the findings in their manuscript fully available?

Reviewer #1: No

Reviewer #2: No

4. Is the manuscript presented in an intelligible fashion and written in standard English?

Reviewer #1: Yes

Reviewer #2: Yes

5. Review Comments to the Author

Reviewer #1: See attached file, manuscript needs major corrections and can be acceptable after revisions. Major issues are related to text flow, limited intro, missing discussions, and conclusions.

Major/minor issues

Ln24-27, do you have obs, why only WRF used?

Ln48; Gultepe et al AMS bulletin see for this impact

Ln49-50; that is not correct, if not validated

LN50; advection cooling, not good term

Ln52; same issue adv cooling?

Ln62; Gultepe et al 2021, lately studied marine fog from CFOG project suggested various important processes play an important role for marine fog formation.

Introduction is very limited, in fact many work done in this respect, see Gultepe et al review papers (see below), nothing mentioned.

Ln64; all know this and LN 66 most severe of what?

Ln83; what are the specific conditions are looked for? SST important all know but what is your contribution.

LN121-130; where are the observations, and how they are used?

Section 3.2 Observations; should be given first, not after model story.

Section 3.4 Comparison of simulated sea fog?? With what? Should say “verification of simulations”NEED

New Section is needed: DISCUSSIONS; I DON’T SEE.

Conclusions

Ln 326; how much properly simulated, observations were not clearly presented.

You need to show first obs and then make conclusions, if no observations then you should say it in the start of paper. But later you say ASOS observations and satellite data.

Manuscript needs in serious improvements, please also see review papers on marine fog that needs to be mentioned:

Suggested papers:

Yang, D., H. Ritchie, S. Desjardins, G. Pearson, A. MacAfee, and I. Gultepe, 2009: High Resolution GEM-LAM application in marine fog prediction: Evaluation and diagnosis. Weather and Forecasting. V. 25, 727-748.

Gultepe, I., Milbrandt, J. A., and Zhou, B., 2017: Marine fog: A review on microphysics and visibility prediction. A chapter In the book of Marine Fog: Challenges and Advancements in Observations, Modeling, and Forecasting, Springer Pub. Comp. NY 10012 USA. Edited by Darko Koracin and Clive Dorman. 345-394 pp.

Gultepe, I., A.J. Heymsfield, H.J.S Fernando, E. Pardyjak, C. E. Dorman, Q. Wang, E. Creegan, S. W. Hoch,, D. D. Flagg, R.Yamaguchi, R. Krishnamurthy, S. Gaberšek, W. Perrie, A. Perelet, D.K. Singh, R. Chang, B. Nagare, S. Wagh, and S. Wang, 2021: A review of Coastal Fog Microphysics during C-FOG. Bound. Layer. Meteor.

Gultepe, I., G. Pearson J. A. Milbrandt, B. Hansen, S. Platnick, P. Taylor, M. Gordon, J. P. Oakley, and S.G. Cober, 2009: The fog remote sensing and modeling (FRAM) field project. Bull. Of Amer. Meteor. Soc., v.90, 341-359.

Reviewer #2: 1. Summary

Because most sea fogs belong to advective fog, the difference between AT and SST ia a vital factor for sea fog formation. Warm-mosit air mass advection and cool sea surface help to make this difference. High-resolution SST can capture more details of coastal sea fog than low-resolution SST, it's not surprise and not a new finding. Therefore, the novelty of this article is not enough.

2. major comments

1) Observed facts

a) the phenomenon of sea fog occurence due to upwelling cold water

Please make a statistics of the spatio-temporal distribution of sea fog along the Korean coast. It's a neccessary evidence for your research motivation.

b) the observed sea fog in the case study

Please show the observed sea fog (modis satellite pciture, radio soudings/CLIPSO for juding the depth of sea fog ), which is used to verify/evaluate the simulated sea fog.

Visit https://worldview.earthdata.nasa.gov for modis pictures.

2) Sensitivity experiments

Table 3 shows the differeces bwtween the observed SST and the reanalysis SSTs. Even for the high-resolution RTG SST, the dieffrence can reach 2K difference and even more. So, suggest to conduct SST sensitivity experments.

3) Veritical resolution of the WRF simulation

Details of eta-levels ii not seen in the article. For sea fog simulation, vertical resolution near sea surface is very vital.

4) More cases

One case study is not enough.

3. Minor comments

1) The number and title is not seen below the figure in PDF manuscript. It's very inconvenient for reading.

2) In the last figure, "cloud water mixign ratio" and "water vapor mixing ratio" are upside down.

6. PLOS authors have the option to publish the peer review history of their article (what does this mean?). If published, this will include your full peer review and any attached files.

Reviewer #1: No

Reviewer #2: No

---

## [Author Response · Author response to Decision Letter 0]

23 Dec 2021

We have submitted the responses with separated documents via attachment files.

Please check our responses through those files.

---

## [Decision Letter · Decision Letter 1]

19 Apr 2022

Numerical study on advective fog formation and its characteristic associated with cold water upwelling

PONE-D-21-26199R1

Dear Dr. Lee,

We’re pleased to inform you that your manuscript has been judged scientifically suitable for publication and will be formally accepted for publication once it meets all outstanding technical requirements.

We are sorry for the delay in processing the revised version of the manuscript, which was due to the unavailability of the previous Reviewer 2, and the difficulties in finding an alternative reviewer.

Kind regards,

Andrea Storto

Academic Editor

PLOS ONE

Additional Editor Comments (optional):

Reviewers' comments:

Reviewer's Responses to Questions

**Comments to the Author**

1. If the authors have adequately addressed your comments raised in a previous round of review and you feel that this manuscript is now acceptable for publication, you may indicate that here to bypass the “Comments to the Author” section, enter your conflict of interest statement in the “Confidential to Editor” section, and submit your "Accept" recommendation.

Reviewer #1: All comments have been addressed

Reviewer #3: All comments have been addressed

2. Is the manuscript technically sound, and do the data support the conclusions?

Reviewer #1: Yes

Reviewer #3: Partly

3. Has the statistical analysis been performed appropriately and rigorously? 

Reviewer #1: Yes

Reviewer #3: No

4. Have the authors made all data underlying the findings in their manuscript fully available?

Reviewer #1: Yes

Reviewer #3: Yes

5. Is the manuscript presented in an intelligible fashion and written in standard English?

Reviewer #1: Yes

Reviewer #3: Yes

6. Review Comments to the Author

Reviewer #1: Corrections done are satisfactory and manuscript is improved significantly. Authors followed up scientific criteria and properly improved text flow. Your manuscript can now be accepted.

Reviewer #3: (No Response)

7. PLOS authors have the option to publish the peer review history of their article (what does this mean?). If published, this will include your full peer review and any attached files.

Reviewer #1: **Yes: **Ismail Gultepe

Reviewer #3: No

---

## [Editor Report · Acceptance letter]

28 Jul 2022

PONE-D-21-26199R1 

Numerical study on advective fog formation and its characteristic associated with cold water upwelling 

Dear Dr. Lee:

I'm pleased to inform you that your manuscript has been deemed suitable for publication in PLOS ONE. Congratulations! Your manuscript is now with our production department. 

Kind regards, 

on behalf of

Dr. Andrea Storto 

Academic Editor

PLOS ONE